# Avicularin Attenuated Lead-Induced Ferroptosis, Neuroinflammation, and Memory Impairment in Mice

**DOI:** 10.3390/antiox13081024

**Published:** 2024-08-22

**Authors:** Jun-Tao Guo, Chao Cheng, Jia-Xue Shi, Wen-Ting Zhang, Han Sun, Chan-Min Liu

**Affiliations:** School of Life Science, Jiangsu Normal University, No. 101, Shanghai Road, Tongshan New Area, Xuzhou 221116, China; 15399550275@163.com (J.-T.G.); chengchao@jsnu.edu.cn (C.C.); 2020221660@jsnu.edu.cn (J.-X.S.); 20171002@jaas.ac.cn (W.-T.Z.); hanhan-hanhan@126.com (H.S.)

**Keywords:** avicularin, lead, memory impairment, ferroptosis, glucose metabolism, inflammation

## Abstract

Lead (Pb) is a common environmental neurotoxicant that results in abnormal neurobehavior and impaired memory. Avicularin (AVL), the main dietary flavonoid found in several plants and fruits, exhibits neuroprotective and hepatoprotective properties. In the present study, the effects of AVL on Pb-induced neurotoxicity were evaluated using ICR mice to investigate the molecular mechanisms behind its protective effects. Our study has demonstrated that AVL treatment significantly ameliorated memory impairment induced by lead (Pb). Furthermore, AVL mitigated Pb-triggered neuroinflammation, ferroptosis, and oxidative stress. The inhibition of Pb-induced oxidative stress in the brain by AVL was evidenced by the reduction in malondialdehyde (MDA) levels and the enhancement of glutathione (GSH) and glutathione peroxidase (GPx) activities. Additionally, in the context of lead-induced neurotoxicity, AVL mitigated ferroptosis by increasing the expression of GPX4 and reducing ferrous iron levels (Fe^2+^). AVL increased the activities of glycogenolysis rate-limiting enzymes HK, PK, and PYG. Additionally, AVL downregulated TNF-α and IL-1β expression while concurrently enhancing the activations of AMPK, Nrf2, HO-1, NQO1, PSD-95, SNAP-25, CaMKII, and CREB in the brains of mice. The findings from this study suggest that AVL mitigates the memory impairment induced by Pb, which is associated with the AMPK/Nrf2 pathway and ferroptosis.

## 1. Introduction

Lead (Pb) is a pervasive neurotoxic contaminant that poses a significant global health concern, especially in relation to neurodevelopmental disorders [1,2,3,4]. Exposure to Pb has been found to disrupt brain glucose metabolism and after the expression levels of crucial regulatory enzymes such as hexokinase (HK), pyruvate kinase (PK), and glycogen phosphorylase (PYG) [2,3]. Prior research has demonstrated that exposure to Pb can result in an increase in abnormal neurobehavior, impaired memory, inflammation, oxidative stress, and autophagy. These effects are associated with the regulation of AMP-activated protein kinase (AMPK), a universally conserved sensor of cellular energy alterations [5,6]. Pb also causes neurotoxicity, which is closely related to oxidative stress, inflammation and cell death by regulating the Nrf2/ARE pathway [7,8].

Ferroptosis, a distinct form of cell death, is intimately linked to iron-dependent lipid peroxidation and is characterized by the accumulation of iron and malondialdehyde (MDA), as well as lipid peroxidation and glutathione (GSH) depletion [9]. The enzyme glutathione peroxidases 4 (GPX4) plays a pivotal role in the regulation of ferroptosis. GPX4 mitigates ferroptosis by decreasing lipid peroxidation, a process that necessitates the involvement of GSH [10,11]. Previous evidence has demonstrated that Pb induces neuronal ferroptosis through the disruption of iron metabolism and the promotion of oxidative stress [9,10]. Recent findings further substantiated that Pb exposure is associated with cognitive dysfunction and hippocampal ferroptosis [11]. Pb is associated with an abnormal impairment of memory, which is linked to the regulation of synaptosomal associated protein-25 (SNAP-25), postsynaptic density-95 (PSD-95), cyclic-AMP-response element-binding protein (CREB), and calcium/calmodulin kinase II (CaMKII) [5,12,13].

Avicularin (AVL, quercetin-3-alpha-L-arabinofuranoside), a glycoside of quercetin found in several plants and fruits, has displayed neuroprotective and hepatoprotective properties [14,15,16]. AVL exhibits potential as an anti-inflammatory agent in the treatment of hepatic disorders and neurodegenerative diseases through its ability to inhibit the production of inflammatory factors such as TNF-α and Interleukin-beta (IL-1β) [14,15]. Research has demonstrated that the supplementation of AVL has the potential to enhance cognitive activity in behavioral studies. Furthermore, it has been observed to counteract the inflammatory response and excessive oxidative stress in rats afflicted with Amyloid Beta-Induced Alzheimer’s Disease [15]. Supplementation with AVL ameliorated the insulin resistance linked to the regulation of expression levels of PYG and PK [16,17]. Moreover, AVL enhances glucose metabolism through the augmentation of HK and PK activities [17,18].

Therefore, AVL was investigated for its neuroprotective role against Pb-induced synaptic dysfunction, ferroptosis, inflammation and memory impairment, and to clarify the role of the AMPK/Nrf2 pathway in AVL protection.

## 2. Materials and Methods

### 2.1. Chemicals and Reagents

Avicularin (purity > 99%) and lead acetate were purchased from Sigma Chemical Co. (St. Louis, MO, USA). AMPK (ab80039, Abcam, Cambridge, UK), p-AMPK (ab133448), PYG (ab154969), HK2 (ab209847), PKM2 (ab194112,) GPX4 (ab125066, Abcam), IL-1β (ab254360,Abcam), TNF-α (ab183218, Abcam), SNAP-25 (ab2192054), PSD-95 (ab18258), p-CREB (ab32096, Abcam), CaMKII (ab11959), CREB (ab32515, Abcam), Nrf2 (ab31163, Abcam), HO-1 (ab68477, Abcam), NQO1 (ab28947, Abcam), and GAPDH (ab8245, Abcam) antibodies were provided by Abcam (Cambridge, MA, USA) [5,18].

### 2.2. Experimental Design

Fifty 8-week-old male ICR mice were obtained from Jinan Experimental Animal Co., Ltd. (Jiannan, China). They were given ample water, adequate food, and maintained on a 12 h light/dark cycle per Laboratory Animals Guidelines. After one week of acclimatization, the mice were randomly divided into five groups of ten: (1) Normal control; (2) Pb; (3) Pb + AVL (25 mg/kg); (4) Pb + AVL (50 mg/kg); (5) AVL (50 mg/kg) control. Groups (2), (3), and (4) underwent Pb-induced nerve damage using a 1000 mg/L lead acetate solution in drinking water for 3 months [11]. Additionally, groups (3), (4), and (5) received AVL at 25 or 50 mg/kg intragastrically once daily [15,16]. A flowchart of the experiment is shown in Figure 1.

### 2.3. Behavioral Tests

As per our previous report, the learning and memory behavior was assessed using a step-down test [5]. Briefly, the step-down test training occurred 24 h after the final AVL dose to evaluate stimulus avoidance. The mice were placed on a copper grid and given an electric shock. Over a 3 min session, the time taken for each mouse to first jump off the platform was recorded, along with the number of shocks received, noted as errors. After a 24 h period, the test was administered again to assess memory, with the latency and number of errors being recorded as memory test scores. 

An object recognition task was used to measure hippocampal-dependent recognition memory behavior [19]. Briefly, during the acquisition phase of the experiment, the mice were positioned in the center of the arena and given the opportunity to investigate two identical objects that were equidistant from each other for a duration of 8 min. Following the conclusion of the acquisition phase, the mice were then placed back in their home cage for a 1 h inter-trial interval (ITI). After the ITI, the mice underwent the same conditions as those in the acquisition phase, with the exception of one of the identical objects being substituted with a novel object. Interaction with the object was noted when the mouse sniffed, touched, stood, sat, or leaned on it. The novelty preference index was calculated by dividing the time spent on the novel object by the total interaction time with both objects, and then multiplying it by 100. A novelty preference index above 50% indicated successful task performance.

### 2.4. Biochemical Analysis

To assess biochemical parameters, the homogenates were centrifuged for 30 min at 5000 rpm at 4 °C. The levels of MDA (#A003-1), GSH (#A006-2), SOD (#A003-3-1), and GPx (#A005-1-2) in the brains were analyzed by using commercial kits (Jiancheng Institute of Biotechnology, Nanjing, China) [18]. GSH reacts with DTNB to produce 2-nitro-5-mercaptobenzoic acid and GSSG. The yellow color of the produced acid, whose intensity correlates with GSH concentration, allows for GSH content determination by measuring its absorbance at 412 nm. GPx catalyzes GSH oxidation into GSSG using hydrogen peroxide. GSH also reacts with 5,5’-dithiobis-(2-nitrobenzoic acid) to form a yellow compound with a 412 nm absorption peak. The enzyme’s activity is determined by the absorbance decrease rate. Ferrous iron levels were measured using an iron assay kit (ab83366, Abcam) [11]. Free Fe^2+^ reacts with Iron Probe to form a colored complex, and absorbance at 593 nm was recorded using a microplate reader.

### 2.5. Immunofluorescence

Immunofluorescence analysis was conducted as previously described [20,21]. Brain tissues were fixed in 4% paraformaldehyde for 24–48 h, dehydrated in ethanol, and embedded in paraffin. Sections (5 μm thick) were deparaffinized, underwent antigen retrieval, and were blocked with 10% fetal bovine serum in PBS for 40 min. They were then incubated overnight at 4 °C with primary antibodies anti-GPx4 (1:200; ab125066; Abcam) and anti-NeuN (1:200; ab104224; Abcam). Sections were washed, incubated with fluorescein-labeled secondary antibodies (1:200; Abbkine, Wuhan, China), and imaged using a Carl Zeiss fluorescence microscope (Oberkochen, Germany).

### 2.6. Western Blot Analysis

Nuclear and cytoplasmic extracts for Western blotting were obtained by using a nuclear/cytoplasmic isolation kit (Beyotime Institute of Biotechnology, Beijing, China). Molecular masses were determined using stained protein markers (PR1910, Beijing Solarbio Science & Technology Co., Ltd., Beijing, China). Hippocampal tissues were homogenized in 3 mL of ice-cold RIPA lysis buffer with added PMSF, Na_3_VO_4_, and protease inhibitors. The homogenates were sonicated, centrifuged twice, and the supernatants were collected. Protein levels in the supernatants were measured using the BCA assay kit (Pierce Biotechnology, Inc., Rockford, IL, USA). Samples (60 g) were separated by SDS-PAGE and transferred to a PVDF membrane using electrophoretic transfer. The membrane was pre-blocked with 5% non-fat milk and 0.1% Tween-20 in TBST, and then incubated overnight with the primary antibody in the same solution. After three 15 min washes, the membrane was incubated with secondary HRP-linked antibodies (Abcam, USA). Blot densities were normalized to GAPDH and averaged from three samples [18]. The vehicle control density was set to 1.0 for relative comparison with other groups. Analysis was carried out using Image J 1.42 software (NIH, Bethesda, MD, USA).

### 2.7. Statistical Analysis

The data were analyzed using one-way ANOVA with a Tukey post hoc test. The results are presented as mean + SEM. The significance level was set at *p* < 0.05.

## 3. Results

### 3.1. AVL Rescues Pb-Induced Impairment of Learning and Memory

Exposure to Pb led to a notable decline in learning and memory abilities in comparison to the control group. Specifically, Pb exposure was associated with decreased latency in both learning and memory tasks (by 34.44% and 35.18%, respectively) and an increase in the number of errors made during these tasks (by 83.27% and 265.11%, respectively) when compared to the control group. The administration of AVL resulted in a reduction in latency and a decrease in the number of errors when compared to the Pb group (Table 1). However, treatment with AVL alone did not yield any statistically significant effects on behavior.

### 3.2. AVL Rescues Pb-Induced Impairment of Novel Object Recognition Memory

Exposure to lead (Pb) significantly impaired novel object recognition compared to the control group, as evidenced by a decrease in object exploration during the acquisition phase by 42.72%. Additionally, mice exposed to Pb showed a reduced preference index for the novel object by 38.52% compared to vehicle-treated controls. Conversely, the administration of AVL increased total exploration time and preference index for the novel object compared to the Pb group, indicating a potential impairment in recognition memory in the Pb-exposed group. Furthermore, treatment with AVL appeared to reverse this deficit (Table 2).

### 3.3. Effects of AVL on Pb-Induced Ferroptosis in Brains

To investigate the therapeutic efficacy of AVL in mitigating Pb-induced ferroptosis, the levels of Fe^2+^ in brain tissues were assessed (Figure 2). Exposure to Pb resulted in a 213.64% elevation in Fe²⁺ concentrations within the brain tissue relative to the control group. The administration of AVL at dosages of 25 mg/kg and 50 mg/kg resulted in a reduction in Fe^2+^ levels by 18.12% and 32.97%, respectively. An evaluation of GPX4 expression was performed through immunofluorescence and western blot assays. Pb exposure markedly decreased the prevalence of GPX4-positive neurons in the immunofluorescence staining of the cerebral cortex. Conversely, AVL treatment increased GPX4-positive neurons in the brain (Figure 2A). The results of the western blot analysis demonstrated a significant decrease in GPX4 expression by 71% following Pb exposure compared to the control group. In contrast, AVL treatment increased GPX4 expression in brains compared to the Pb-exposed group (Figure 2B).

### 3.4. AVL Inhibits Pb-Induced Oxidative Stress in Brains

In order to assess the impact of AVL on Pb-induced oxidative stress in the brain, the levels of GSH and MDA, as well as the GPX activities, were measured. Pb exposure resulted in a decrease in GSH levels by 55.92%, as well as a decrease in the GPX activities by 30.63%, while also leading to an increase in MDA levels by 109.82% compared to the normal control group. Conversely, AVL treatment boosted antioxidant enzyme activities and reduced lipid peroxidation (Figure 3).

### 3.5. AVL Regulated the Expression Levels of Glucose Metabolism Enzymes in Brains

Western blot analysis was utilized to evaluate the expression levels of enzymes involved in glucose metabolism. As illustrated in Figure 4, the data indicate that exposure to Pb resulted in a significant decrease in the protein levels of HK2, PKM2, and PYG when compared to the control group. Conversely, treatment with AVL restored the protein expression levels of these glucose metabolism enzymes.

### 3.6. AVL Suppressed Neuroinflammation

Inflammation is a significant factor in the development of nerve damage. This investigation assessed the concentrations of pro-inflammatory cytokines TNF-α and IL-1β within cerebral tissue. As illustrated in Figure 5, lead (Pb) exposure was associated with a significant increase in the levels of TNF-α and IL-1β relative to the control group. AVL treatment reduced the expression of inflammatory factors compared to the Pb-exposed group.

### 3.7. Effect of AVL on the AMPK/Nrf2 Signaling Pathway

This study conducted a detailed analysis of the expression levels of AMPK, Nrf2, HO-1, and NQO1. The results illustrated in Figure 6 indicate that the levels of p-AMPK, Nrf2, HO-1, and NQO1 were decreased in the Pb group. Conversely, the administration of varying doses of AVL notably elevated the expression of these proteins compared to the Pb group.

### 3.8. AVL Attenuated Pb-Induced Neurotoxicity

To assess the impact of AVL treatments on lead-induced neurotoxicity, the markers of nerve damage SNAP-25, PSD-95, CREB, and CaMKII were analyzed. The results depicted in Figure 7 indicate a decrease in the levels of SNAP-25, PSD-95, phosphorylated CREB (p-CREB), and phosphorylated CaMKII (p-CaMKII) following Pb exposure. Conversely, the administration of AVI was found to elevate the expression levels of these proteins.

## 4. Discussion

Lead (Pb) is a widely distributed, enduring, and non-essential toxic heavy metal capable of causing nerve damage [7,8]. Avicularin, a plant-derived flavonoid and a glycoside of quercetin, exhibits a wide range of pharmacological activities, encompassing antioxidative, anti-inflammatory, anticancer, antiallergic, antidepressive, and hypoglycemic effects [14,15,16,17,18]. Numerous studies have demonstrated that exposure to Pb can lead to neurotoxic and neurobehavioral effects, resulting in memory impairments and synaptic dysfunction [1,5,7]. Our research identified abnormal memory behavior as a consequence of Pb exposure (Table 1 and Table 2). Additionally, we observed that supplementation with AVL mitigated the nerve injury induced by Pb. These findings align with previous research indicating the protective effects of AVL against Aβ-induced Alzheimer’s disease [15].

Ferroptosis is a non-canonical form of programmed cell death distinguished by iron-dependent lipid peroxidation, resulting in the depletion of glutathione and the accumulation of unmetabolized lipid peroxides. The oxidation of lipids by Fe^2+^ generates ROS, thereby facilitating the progression of ferroptosis. Biochemically, ferroptosis is characterized by elevated levels of ROS, Fe^2+^, and MDA, along with diminished activity of antioxidant enzymes like GPX4 and GSH [22,23]. This mode of cell death is implicated in various neurological disorders [24]. Pb exposure has been shown to enhance the buildup of Fe^2+^ and ferroptosis in brain tissues and nerve cells [9,10,11]. AVL has demonstrated the ability to mitigate LPS-induced accumulation of Fe^2+^ and ferroptosis in the liver [10]. Additionally, the depletion of GSH/GPX4 and the subsequent increase in lipid peroxidation are characteristic features of ferroptosis [10,24,25]. Glutathione serves as a crucial antioxidant and scavenger of free radicals in the body. It has the ability to bind with free radicals and heavy metals, transforming harmful toxins into benign substances that can be eliminated from the body. Additionally, glutathione plays a dual role in combating abnormal free radicals and supporting the function of GPX4, the primary catalytic enzyme involved in ferroptosis [25]. GPX4 is essential for regulating ferroptosis by balancing glutathione levels and catalyzing the reduction in lipid peroxidation [24,26]. The results of the current study indicate that exposure to Pb results in an increase in Fe^2+^ accumulation and down-regulation of GPX4, potentially leading to an increase in ferroptosis. Previous research has also suggested that Pb exposure can induce ferroptosis and oxidative damage in the brain (Figure 2). Consistent with these findings, our study observed an increase in ferroptosis and oxidative stress in mouse brains following Pb exposure, as evidenced by elevated MDA concentration and decreased activities of GSH and GPx (Figure 3). Prior studies have shown that AVL supplementation inhibited ferroptosis and oxidative damage in mouse livers [10]. Our research demonstrates that AVL treatment inhibited Pb-induced ferroptosis and oxidative stress in brain tissue, suggesting its potential utility as a neuroprotective agent.

Ferroptosis plays a role in modulating cellular energy metabolism by sensitizing cells to elevated levels of ROS and disrupting iron homeostasis when glycolysis is inhibited [22,27]. Studies have shown that ferroptosis decreases glycolytic activity by downregulating essential glycolysis enzymes such as HK2 and PKM2 [28,29]. Exposure to lead can disrupt glucose metabolism and contribute to the development of diabetes in animal models [2,19,30] Treatment with AVL has been shown to ameliorate dysfunction in glucose metabolism in a type 2 diabetes model [16,17]. PYG serves as the rate-limiting enzyme in glycogenolysis, while PKM and HK are crucial glycolytic enzymes that regulate the rate of glycolysis and are implicated in various neurological disorders [16,31,32]. Studies have demonstrated that Pb exposure reduces the expression of HK, PK, and PYG in brain and liver tissues [2,3,30]. Conversely, research has shown that AVL can increase the expression of PK and PYG, thereby improving glucose metabolism dysfunction [16]. Our current investigation revealed that AVL treatment restored the activities of HK2 and PYG in the brains of Pb-exposed subjects, suggesting that AVL mitigates Pb-induced glucose metabolism disturbances (Figure 4).

The phenomenon of ferroptosis has been linked to the facilitation of inflammation, with ferroptosis inhibitors demonstrating antioxidant and anti-inflammatory characteristics [33]. The inflammatory response is closely correlated with glucose metabolism, as evidenced by the stimulation of inflammation and disruption of gluconeogenesis and glycogenolysis in the liver following exposure to lead [21,30]. Multiple studies have demonstrated that AVL has the ability to suppress the formation of inflammasome in the liver and brain across various experimental models, thereby mitigating tissue damage caused by toxins [14,34]. AVL has also been shown to inhibit the release of pro-inflammatory cytokines IL-1β, IL-6, and TNF-α in different cell types [35,36,37]. The findings of this investigation indicate that Pb induces the secretion of IL-1β and TNF-α, whereas AVL effectively inhibits the production of these inflammatory mediators (Figure 5), suggesting a potential therapeutic role for AVL in mitigating brain injury by attenuating Pb-induced inflammation.

AMPK is recognized as a conserved cellular energy sensor [5,6]. The activation of AMPK during periods of energy stress has been shown to decrease the production of fatty acids susceptible to ferroptosis while also preserving antioxidative defenses [11,22]. Furthermore, activated AMPK demonstrates resistance to ferroptosis, whereas the inactivation of AMPK renders cells more susceptible to ferroptosis. Additionally, activated AMPK has been found to inhibit ferroptosis, oxidative stress, and inflammatory responses [38,39,40]. Nrf2 has been demonstrated to exhibit a strong correlation with lipid peroxidation and ferroptosis. Upon translocation, Nrf2 facilitates the transcription of downstream genes such as HO-1, NQO1, and GPX4, whose corresponding proteins are pivotal in antioxidation and anti-ferroptosis mechanisms [11,41]. Furthermore, research has demonstrated that the activation of AMPK can mitigate lipid peroxidation and ferroptosis in neurodegenerative disorders, a process associated with the modulation of the Nrf2 signaling pathway [11,38]. Many studies have shown that Pb can instigate oxidative stress, inflammation, and neurotoxicity by inhibiting the AMPK signaling pathways [5,42]. Furthermore, exposure to Pb can result in neuroinflammation and cerebellar impairments, phenomena associated with the inhibition of the Nrf2 antioxidant pathway [7,8]. However, the administration of AVL has been found to mitigate oxidative stress, inflammatory response, and ferroptosis, processes closely tied to the activation of the Nrf2/HO-1/GPX4 pathway [10]. Our study revealed that Pb reduces the expression of p-AMPK and Nrf2, while AVL supplementation restores the activation of these proteins in the brain (Figure 6). Therefore, it can be concluded that AVL has the potential to mitigate Pb-induced brain damage by activating the AMPK/Nrf2/GPX4 pathway.

The presynaptic protein SNAP-25, postsynaptic protein PSD-95, transcription factor CREB, and CAMKII are crucial components in neuronal development, plasticity, and cognitive/synaptic function [5,12]. Numerous studies have demonstrated that exposure to lead results in cognitive/synaptic impairments and reduced expression of SNAP-25, PSD-95, CREB, and CAMKII in the brain [5,12,43]. A prior investigation demonstrated that AVL possesses neuroprotective qualities through the amelioration of memory deficits [15,34]. The findings of the present study indicate that Pb diminishes the expression of SNAP-25, PSD-95, p-CREB, and p-CAMKII in the brain. Conversely, AVL significantly enhances the expression of these proteins (Figure 7), suggesting that AVL mitigates brain injury by regulating synaptic and memory-associated proteins.

In summary, the activation of the AMPK/Nrf2 pathway by AVL was associated with notable reductions in Pb-induced memory impairment, inflammation, ferroptosis, oxidative stress, and glucose metabolism disorder (Figure 8). Avicularin has the potential to serve as a promising neuroprotective agent. While this research noted that AVL could mitigate the neurotoxic effects induced by Pb in mice, the precise underlying mechanism remains ambiguous. Further exploration is warranted to understand the influence of AVL on other pivotal components involved in the pathogenesis of Pb-induced neurotoxicity.

## Figures and Tables

**Figure 1 antioxidants-13-01024-f001:**
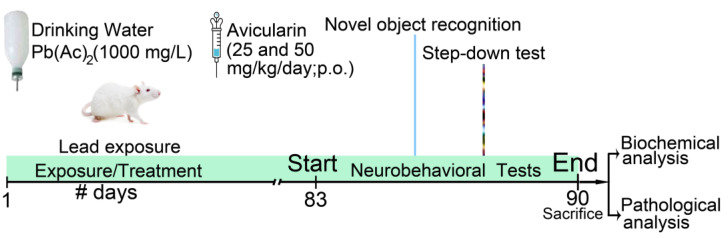
The schematic representation of timeline for the experiments.

**Figure 2 antioxidants-13-01024-f002:**
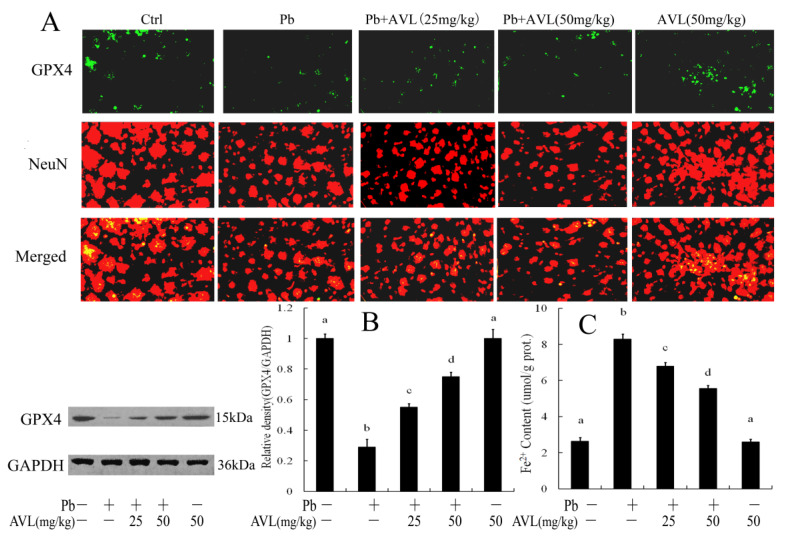
Avicularin (AVL) inhibited Pb-induced ferroptosis in the brains of mice. (**A**) Immunofluorescence analysis of ferroptosis in the cerebral cortex (200×) (GPX4, green; NeuN, red; Merged, yellow); (**B**) Western blot analysis of the GPX4 proteins in the brains. (**C**) Fe^2+^ levels in the brains. Data are expressed as mean ± SEM (*n* = 3). Values not sharing a common superscript letter (a–d) differ significantly at *p* < 0.05.

**Figure 3 antioxidants-13-01024-f003:**
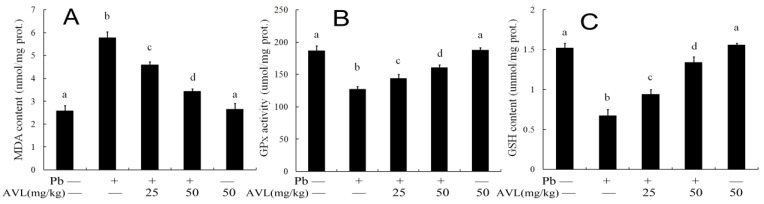
Effect of avicularin on Pb-induced oxidative stress in brains of mice. (**A**) MDA content; (**B**) GPx activity; (**C**) content. Data are expressed as mean ± S.E.M (*n* = 7). Values not sharing a common superscript letter (a–d) differ significantly at *p* < 0.05.

**Figure 4 antioxidants-13-01024-f004:**
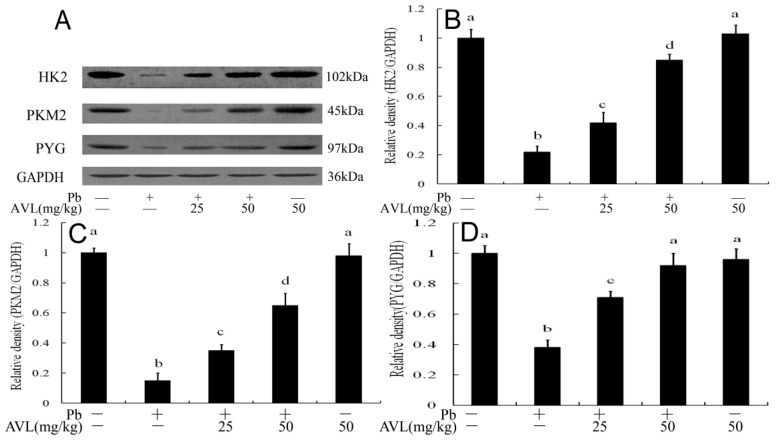
The protein expression of the glucose metabolism in the brains of mice. (**A**) Western blotting was used to assess the expression levels of glucose metabolism-related proteins; (**B**–**D**) HK2, PKM2 and PYG protein quantification. Data are expressed as mean ± SEM (*n* = 3). Values not sharing a common superscript letter (a–d) differ significantly at *p* < 0.05.

**Figure 5 antioxidants-13-01024-f005:**
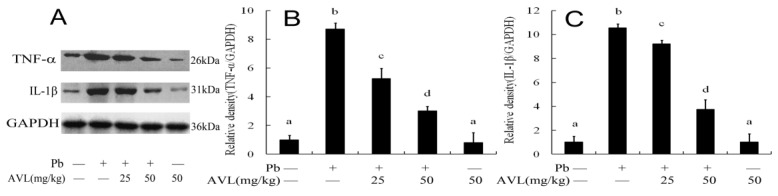
Avicularin (AVL) inhibited Pb-induced inflammation in the brains of mice. (**A**) Western blotting was used to assess the expression levels of inflammatory cytokines; (**B**,**C**) TNF-α and IL-1β protein quantification. Data are expressed as mean ± SEM (*n* = 3). Values not sharing a common superscript letter (a–d) differ significantly at *p* < 0.05.

**Figure 6 antioxidants-13-01024-f006:**
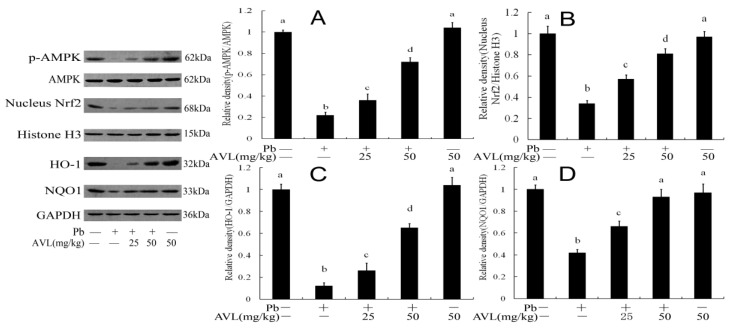
Avicularin (AVL) activated the AMPK/Nrf2 pathway in the brains of mice. (**A**–**D**) p-AMPK, nucleus Nrf2, HO-1, and NQO1 protein quantification. Data are expressed as mean ± SEM (*n* = 3). Values not sharing a common superscript letter (a–d) differ significantly at *p* < 0.05.

**Figure 7 antioxidants-13-01024-f007:**
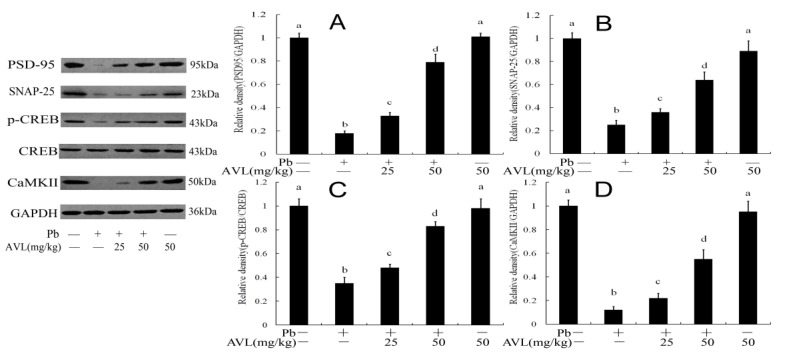
Avicularin (AVL) activated synaptic regulation pathway in the brains of mice. (**A**–**D**) PSD-95, SNAP-25, p-CREB, and CaMKII protein quantification. Data are expressed as mean ± SEM (*n* = 3). Values not sharing a common superscript letter (a–d) differ significantly at *p* < 0.05.

**Figure 8 antioxidants-13-01024-f008:**
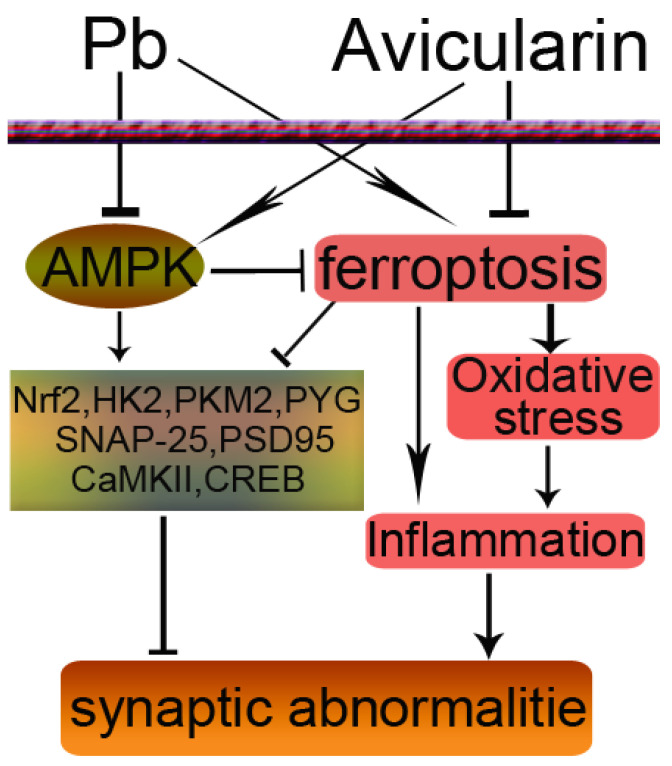
Schematic diagram showing the possible protective effects of avicularin (AVL) in Pb-induced brain injury. The → indicates activation or induction, and ┤ indicates inhibition or blockade.

**Table 1 antioxidants-13-01024-t001:** Effect of avicularin on Pb-induced impairment of memory in mice.

	Latency (Second)		The Number of Errors	
	Learning Training Test	Memory Test	Learning Training Test	Memory Test
Control	78.95 ± 3.92 ^a^	103.46 ± 3.30 ^a^	2.57 ± 0.50 ^a^	0.43 ± 0.47 ^a^
Pb	53.34 ± 3.63 ^b^	67.06 ± 2.53 ^b^	4.71 ± 0.47 ^b^	1.57 ± 0.47 ^b^
Pb + AVL (25 mg/kg)	69.67 ± 2.89 ^c^	81.11 ± 1.65 ^c^	3.29 ± 0.47 ^c^	1.14 ± 0.58 ^c^
Pb + AVL (50 mg/kg)	74.71 ± 1.89 ^d^	89.78 ± 1.78 ^d^	2.86 ± 0.37 ^d^	1.00 ± 0.37 ^d^
AVL (50 mg/kg)	78.70 ± 2.19 ^a^	103.85 ± 3.19 ^a^	2.29 ± 0.47 ^a^	0.29 ± 0.47 ^a^

Data are expressed as mean ± SEM. (*n* = 7). Values not sharing a common superscript letter (a–d) differ significantly at *p* < 0.05.

**Table 2 antioxidants-13-01024-t002:** Effect of avicularin on Pb-induced impairment of novel object recognition memory in mice.

	Total Exploration Time (Seconds)	Preference Index (%)
Control	30.36 ± 1.15 ^a^	84.13 ± 2.28 ^a^
Pb	17.39 ± 0.88 ^b^	51.72 ± 2.16 ^b^
Pb + AVL (25 mg/kg)	24.27 ± 0.66 ^c^	64.09 ± 1.31 ^c^
Pb + AVL (50 mg/kg)	29.24 ± 0.90 ^a^	71.14 ± 1.13 ^d^
AVL (50 mg/kg)	32.85 ± 2.68 ^a^	85.24 ± 2.61 ^a^

Data are expressed as mean ± SEM. (*n* = 7). Values not sharing a common superscript letter (a–d) differ significantly at *p* < 0.05.

## Data Availability

The datasets generated during and/or analyzed during the current study are available from the corresponding author on reasonable request.

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
