# Peer review of "Avicularin Attenuated Lead-Induced Ferroptosis, Neuroinflammation, and Memory Impairment in Mice"

_antioxidants, 2024, doi:10.3390/antiox13081024_

Round 1

Reviewer 1 Report

1. Although the manuscript presents some novel findings, the conclusions are unsupported by the experimental data.

2. The soundness of the protein expression data presented in the manuscript (Figs. 2b, 3a, 4-6) cannot be properly evaluated because the original Western blot images provided as narrow membrane strips showing single polypeptide bands. Thus, the locations of these bands on the original membranes and the molecular weights of the polypeptides are unknown, which makes it impossible to identify the proteins. This is a major flaw. Therefore, the authors must provide the images of original untrimmed membranes showing molecular weight markers and sample names.  The membrane images should be numbered according to the Figure numbers. 

3. The manuscript presents no direct evidence that avicularin attenuates lead-induced neurotoxicity VIA the AMPK/Nrf2 pathways. Therefore this conclusion is unsupported. Furthermore, the protein expression data are insufficient for the conclusions regarding "glycometabolism" and "synaptic impairment" (see above). 

1. The Title should be revised because it presents inaccurate information and data interpretation (see above)

2. The Abstract does not present a clear and logical overview of the findings. To improve the summary, the description of each of the following effects of AVL - reduced neuroinflammation, ferroptosis, and oxidative stress - needs to be accompanied by the description of the corresponding associated molecular events. 

3. The Introduction needs to be revised, as commented above.

4. Methods: There is no description of sample preparation for different assays.

- Fig. 1: No "histology" or "proteomics" data are presented in this manuscript. 

- The description of biochemical analyses (Section 2.4) should be expanded to indicate the principles of the iron, MDA, GSH, and GPx assays used in the study.

- Section 2.5. describes the immunofluorescence analysis of the LIVER tissue.  

- The Western blot analysis (Section 2.6) must be described in detail.

Results:

The description of the Results is quite brief, without sufficient details. The text also contains incorrect information (e.g., 3.5. AVL regulated the expression levels of glucose metabolism enzymes in LIVER; line 179; or "Conversely, the administration of THREE INHIBITORS was found to elevate..."; line 221) and scientifically incorrect statements (e.g., "In order to substantiate the role of SNAP-25, PSD-95, CREB, and CaMKII in the synaptic abnormalities induced by Pb and the effects of AVL, we conducted an analysis of the protein expression levels"; line 217). The "role" of these proteins was not substantiated by simple Western blot analysis, and the "synaptic abnormalities" were not demonstrated but only speculated upon based on the same protein expression data. 

- Section 3.5. describes protein expression data in the LIVER tissue. 

Technical points:

The manuscript is written in poor scientific English. There a multiple grammar and style errors, wrongly constructed phrases and scientifically incorrect expressions. The entire text should be thoroughly revised by a native English-speaking person or a scientific English editor familiar with this kind of research. 

Author Response

Dear editor,

Thank you very much for your kind considerations on our manuscripts and arranging a timely review for our manuscripts. And we would like to thank referees for critical comments and thoughtful suggestions. We have responded to these suggestions point by point, and revised the manuscript accordingly. All changes made to the text are corrected and indicated by red fonts so that they may be easily identified. Our responses to the reviewers' comments are as follows:

Reviewer #1:

1. The title should be revised because the manuscript presents no direct evidence that avicularin (not “vicularin”) attenuates lead-induced neurotoxicity VIA the AMPK/Nrf2 pathways. Also, no data are provided to support the authors’ conclusion that avicularin upregulates the Nrf2 pathway in the brain through AMPK activation. These conclusions are based solely on correlations between the biological effects of avicularin and changes in AMPK and some proteins related to the Nrf2 pathway. Therefore, the potential involvement of AMPK and/or the Nrf2 pathway may only be considered cautiously in the Discussion but not stated as a proven fact in the title or any other part of the manuscript.

  Response:

 Agree, according to your suggestion, we have revised the title (Page 1).

  1. The Introduction provides a brief overview of previous findings. However, it is too short and offers insufficient background on the processes and regulatory pathways studied in the presented work. For instance, there is no information on the role of iron and GPX-4 in ferroptosis, and thus, it will be unclear to the readers why changes in GPX-4 levels were assessed (Section 3.3). Likewise, the functional roles of SNAP-25 and PSD-95, CREB, and CaMKII are not explained (line 48). Moreover, the relationship between the neurotoxic effects of Pb or neuroprotective effects of avicularin and the molecular events (e.g., protein expression) described in most studies cited in the Introduction was misinterpreted in the current manuscript. Moreover, the authors misinterpreted the findings of most studies cited in the Introduction regarding the links between the neurotoxic effects of Pb or neuroprotective effects of avicularin and the assessed molecular events (e.g., protein expression). Particularly, the cited studies demonstrated only “association” or “correlation” between the above effects and the levels of various proteins (Refs. 2, 3, 5-8, etc.). However, by incorrectly using the expressions containing “VIA”, “BY” or “THROUGH”, the authors made these findings look like there was a causative relationship between the functional and molecular effects of the compounds (lines 40, 43, 47, 51, 55). This can definitely confuse the readers unfamiliar with this kind of research.

  Response:

 Agree. Thanks for your comment. According to your suggestion, we have revised our manuscript (Page 4-5).

  1. The description of the Methods is too brief and contains errors and incorrect information. Detailed comments are presented below.

  Response:

Agree. Thanks for your comment. According to your suggestion, we have revised our manuscript (Page 6-10).

  1. The description of the Results is quite brief, without sufficient details. The text also contains incorrect information (e.g., 3.5. AVL regulated the expression levels of glucose metabolism enzymes in LIVER; line 179; or "Conversely, the administration of THREE INHIBITORS was found to elevate..."; line 221) and scientifically incorrect statements (e.g., "In order to substantiate the role of SNAP-25, PSD-95, CREB, and CaMKII in the synaptic abnormalities induced by Pb and the effects of AVL, we conducted an analysis of the protein expression levels"; line 217). The "role" of these proteins was not substantiated by simple Western blot analysis, and the "synaptic abnormalities" were not demonstrated but only speculated upon based on the same protein expression data.

  Response:

Agree. Thanks for your comment. According to your suggestion, we have revised our manuscript (Page 15,17).

  1. Although the manuscript presents some novel findings, the conclusions are unsupported by the experimental data.

  Response:

Agree. Thanks for your comment. According to your suggestion, we have revised our manuscript. The molecular mechanisms of AVL protective effects against Pb-induced nerve damage are not clear. In our study, effects of AVL on Pb-induced neurotoxicity were evaluated using ICR mice. Our study has demonstrated that AVL treatment significantly ameliorated memory impairment induced by lead (Pb). Furthermore, AVL mitigated Pb-triggered neuroinflammation, ferroptosis, and oxidative stress. We first discovered that AVL inhibited lead-induced ferroptosis and revealed the relevant mechanism.

  1. The soundness of the protein expression data presented in the manuscript (Figs. 2b, 3a, 4-6) cannot be properly evaluated because the original Western blot images provided as narrow membrane strips showing single polypeptide bands. Thus, the locations of these bands on the original membranes and the molecular weights of the polypeptides are unknown, which makes it impossible to identify the proteins. This is a major flaw. Therefore, the authors must provide the images of original untrimmed membranes showing molecular weight markers and sample names. The membrane images should be numbered according to the Figure numbers.

  Response:

Agree. Thanks for your comment. According to your suggestion, we have revised data.  We don't have funding for the big PVDF membrane. We cut the corresponding size of the PVDF membrane for protein translocation based on the molecular weight of the target protein and accorded to the size of the marker protein in protein electrophoresis for protein translocation. We have indicated the molecular weights of the protein bands on the original membranes. The molecular weight markers also showed on the images of original untrimmed membranes. The membrane images had be numbered according to the Figure numbers.

  1. The manuscript presents no direct evidence that avicularin attenuates lead-induced neurotoxicity VIA the AMPK/Nrf2 pathways. Therefore this conclusion is unsupported. Furthermore, the protein expression data are insufficient for the conclusions regarding "glycometabolism" and "synaptic impairment"

  Response:

Agree. Thanks for your comment. According to your suggestion, we have revised the conclusion (Page 23).

  1. The Abstract does not present a clear and logical overview of the findings. To improve the summary, the description of each of the following effects of AVL - reduced neuroinflammation, ferroptosis, and oxidative stress - needs to be accompanied by the description of the corresponding associated molecular events.

  Response:

Agree. Thanks for your comment. According to your suggestion, we have revised the abstract (Page 3).

  1. Fig. 1: No "histology" or "proteomics" data are presented in this manuscript.

  Response:

Agree. Thanks for your comment. According to your suggestion, we have revised the Figure1(Page 7).

  1. The description of biochemical analyses (Section 2.4) should be expanded to indicate the principles of the iron, MDA, GSH, and GPx assays used in the study..

  Response:

Agree. Thanks for your comment. According to your suggestion, we have revised the manuscript (Page 8-9).

  1. Section 2.5. describes the immunofluorescence analysis of the LIVER tissue.

  Response:

Agree. Thanks for your comment. According to your suggestion, we have revised the manuscript (Page 9).

  1. The Western blot analysis (Section 2.6) must be described in detail.

  Response:

Agree. Thanks for your comment. According to your suggestion, we have revised the manuscript (Page 9-11).

  1. The description of the Results is quite brief, without sufficient details. The text also contains incorrect information (e.g., 3.5. AVL regulated the expression levels of glucose metabolism enzymes in LIVER; line 179; or "Conversely, the administration of THREE INHIBITORS was found to elevate..."; line 221) and scientifically incorrect statements (e.g., "In order to substantiate the role of SNAP-25, PSD-95, CREB, and CaMKII in the synaptic abnormalities induced by Pb and the effects of AVL, we conducted an analysis of the protein expression levels"; line 217). The "role" of these proteins was not substantiated by simple Western blot analysis, and the "synaptic abnormalities" were not demonstrated but only speculated upon based on the same protein expression data.

  Response:

Agree. Thanks for your comment. According to your suggestion, we have revised the manuscript (Page 16-18).

  1. 14. Section 3.5. describes protein expression data in the LIVER tissue.

  Response:

Agree. Thanks for your comment. According to your suggestion, we have revised the manuscript (Page 16).

  1. 15. The manuscript is written in poor scientific English. There a multiple grammar and style errors, wrongly constructed phrases and scientifically incorrect expressions. The entire text should be thoroughly revised by a native English-speaking person or a scientific English editor familiar with this kind of research.

  Response:

Agree. Thanks for your comment. According to your suggestion, our paper have revised a native English-speaking person or a scientific English editor familiar with this kind of research.

Essaystar Group

+1-208-975-4235

EssayStar, 93 S Jackson St, Seattle, WA 98104

Thank you very much for the excellent and professional review of our manuscript!

Best regards!

Reviewer 2 Report

In the title it must be written avicularin and not vicularin.

The introduction is weak and must be strongly improved: onfy few information on AVL are provided and the aim of the study is not clearly defined.

The conclusion is inappropriated (only one sentence) "In summary, the activation of the AMPK/Nrf2 pathway by AVL resulted in notable reductions in Pb-induced synaptic impairment, inflammation, ferroptosis, oxidative stress, and glucose metabolism disorder (Fig.7). " and must be improved. More details are required. It is necessary to underline the interest of the study and to present potential prespectives.

Figures 2 and 5 must be improved.

Author Response

Dear editor,

Thank you very much for your kind considerations on our manuscripts and arranging a timely review for our manuscripts. And we would like to thank referees for critical comments and thoughtful suggestions. We have responded to these suggestions point by point, and revised the manuscript accordingly. All changes made to the text are corrected and indicated by red fonts so that they may be easily identified. Our responses to the reviewers' comments are as follows:

Reviewer #2:

  1. The introduction must be strongly improved, mainly the part concerning avicularin (AVL). The present study concerns AVL and only few information are provided on this product. In addition, the aim of the study is not clearly justified and the rational of the study is not clear. What is the reason to focus on lead induced neurotoxicity?

  Response:

    Agree. Thanks for your comment. According to your suggestion, we have revised the introduction section of our manuscript (4-5).

  1. The methods are very briefly described. I consider that more details must be given for all the chapters concerned (even when kits are used).

  Response:

Agree. Thanks for your comment. According to your suggestion, we have revised our manuscript (Page 6-10).

  1. Overall well organized and convincing but must be improved. Figure 2: the immunofluorescence staining is not convincing? Better images must be provided. Figure 5: nucleus NRF2: can you precisely describe how the nucleux are isolated. It is necessary to present data with phospho-NRF2. When NRF2 is activated, phospho-NRF2 is present in the nuclei.

  Response:                                                                                                                    

Agree. Thanks for your comment. According to your suggestion, we have revised our manuscript. We have made every effort to improve image quality. But we don't have good photographic equipment to take high-quality photos. Nuclear and cytoplasmic extracts for Western blotting were obtained by using a nuclear/cytoplasmic isolation kit (Beyotime Institute of Biotechnology, Beijing, China). Nrf2 is active in nuclear transfer, our study detected the nuclear expression level of Nrf2. We found that administration of varying doses of AVL notably elevated the expression of nucleus NRF2 compared to the Pb group.

  1. I think that additionnal information on AVL is required at the beginning of the discussion and that the end of the discussion must be improved.

  Response:

Agree. Thanks for your comment. According to your suggestion, we have revised the introduction section of our manuscript (Page 19).

  1. In the title it must be written avicularin and not vicularin.

  Response:

Agree. Thanks for your comment. According to your suggestion, we have revised our manuscript  title (Page 1).

  1. The introduction is weak and must be strongly improved: onfy few information on AVL are provided and the aim of the study is not clearly defined.

  Response:

Agree. Thanks for your comment. According to your suggestion, we have revised our manuscript (Page 4-5).

  1. The conclusion is inappropriated (only one sentence) "In summary, the activation of the AMPK/Nrf2 pathway by AVL resulted in notable reductions in Pb-induced synaptic impairment, inflammation, ferroptosis, oxidative stress, and glucose metabolism disorder (Fig.7). " and must be improved. More details are required. It is necessary to underline the interest of the study and to present potential prespectives.

  Response:

Agree. Thanks for your comment. According to your suggestion, we have revised our manuscript (Page 24).

  1. Figures 2 and 5 must be improved.

  Response:

Agree. Thanks for your comment. According to your suggestion, we have revised the Figures.

Thank you very much for the excellent and professional review of our manuscript!

Best regards!

Reviewer 3 Report

The article by Jun-Tao Guo, Chao Cheng, Jia-Xue Shi, Wen-Ting Zhang, Han Sun and Chan-Min Liu entitled Vicularin attenuated lead-induced ferroptosis, neuroinflammation, glycometabolism and synaptic impairment via the AMPK/Nrf2 pathways”, propose to investigate the effect of neuroprotective effects against Pb-induced synaptic dysfunction, ferroptosis, inflammation and glycometabolism impairment, and clarify the role of the AMPK/Nrf2 pathway in AVL protection.

 I recommend that the paper be accepted with MAJOR revision:

a)     The authors should check for typographical and grammar error the entire manuscript (space, page lines etc…)

b)    Introduction too short.

c)     What do PB and AVL break up into? d)    What are the behaviors? add description e)     More details on the kits. f)     Western analyzes lack antibodies, concentrations, brands, etc j)      what criteria were used to choose the doses? k)    Please, the figures must all have the same style. l)      Please, check the group name, why in some graphs the experimental group named "d" becomes "a"? m)   Please, add significance in figures n)    What is the limitation of the study? o)    Please, add the paragraph of the discussion. g)    The images of the histological image are over expressed, please, change them.
In which area of ​​the brain are the proteins expressed? h)    for the expression of Mda gsh and gpx better with graph than with table i)      The main markers of ferroptosis are GPX4, GSS, SLC7A11, TFRC, FHC, FLC etc. where are the results?

Author Response

Dear editor,

Thank you very much for your kind considerations on our manuscripts and arranging a timely review for our manuscripts. And we would like to thank referees for critical comments and thoughtful suggestions. We have responded to these suggestions point by point, and revised the manuscript accordingly. All changes made to the text are corrected and indicated by red fonts so that they may be easily identified. Our responses to the reviewers' comments are as follows:

Reviewer: #3

  1. The article by Jun-Tao Guo, Chao Cheng, Jia-Xue Shi, Wen-Ting Zhang, Han Sun and Chan-Min Liu entitled “Vicularin attenuated lead-induced ferroptosis, neuroinflammation, glycometabolism and synaptic impairment via the AMPK/Nrf2 pathways”, propose to investigate the effect of neuroprotective effects against Pb-induced synaptic dysfunction, ferroptosis, inflammation and glycometabolism impairment, and clarify the role of the AMPK/Nrf2 pathway in AVL protection.

  Response:

Agree. Thanks for your comment. Thank you for your affirmation of our article. Thank you for your suggestion.

  1. The authors should check for typographical and grammar error the entire manuscript (space, page lines etc…)

  Response:

Agree. Thanks for your comment. According to your suggestion, we have revised our manuscript.

  1. Introduction too short.

  Response:

Agree. Thanks for your comment. According to your suggestion, we have revised our manuscript (4-5).

  1. What do PB and AVL break up into?

  Response:

Agree. Thanks for your comment. According to your suggestion, we have revised our manuscript. Avicularin (AVL, quercetin-3-alpha-L-arabinofuranoside), is a plant-derived flavonoid and a glycoside of quercetin. In animal, dietary flavonoids (glycoside forms) are enzymatically hydrolyzed and absorbed in the intestine, and are conjugated to their glucuronide/sulfate forms by phase II enzymes in epithelial cells and the liver. Some specific products of bacterial transformation, such as ring-fission products and reduced metabolites, exhibit enhanced properties (Murota et al., 2018).

Murota K., Nakamura Y., Uehara M. Flavonoid metabolism: the interaction of metabolites and gut microbiota. Bioscience, Biotechnology, and Biochemistry, 2018, 82:(4), 600–610.

  1. What are the behaviors? add description.

  Response:

Agree. Thanks for your comment. According to your suggestion, we have revised our manuscript (Page 7).

  1. More details on the kits.

  Response:

Agree. Thanks for your comment. According to your suggestion, we have revised our manuscript. The details on the kits have been clearly described as required (8-10).

7.Western analyzes lack antibodies, concentrations, brands, etc change them. In which area of ​​the brain are the proteins expressed?

  Response:

Agree. Thanks for your comment. According to your suggestion, we have revised our manuscript . The protein expression levels in hippocampal tissues were analyzed by Western blot (8-10).

  1. The images of the histological image are over expressed.

  Response:

Agree. Thanks for your comment. According to your suggestion, we have revised our manuscript. We have made every effort to improve image quality.

  1. The assay of Fe2+ contents in the brain tissues need to add up to support Fig 2.

  Response:

Agree. Thanks for your comment. According to your suggestion, we have revised our manuscript.

  1. for the expression of Mda gsh and gpx better with graph than with table.

  Response:

Agree. Thanks for your comment. According to your suggestion, we have revised our manuscript. We have converted the table into an image (Page 15, Figure 3).

  1. The main markers of ferroptosis are GPX4, GSS, SLC7A11, TFRC, FHC, FLC etc. where are the results?

  Response:

Agree. Thanks for your comment. According to your suggestion, we have revised our manuscript. Ferroptosis is characterized by several specific markers, including phospholipid peroxidation, impaired glutathione peroxidase 4 (GPx4) activity, and Fe2+ accumulation. To investigate the therapeutic efficacy of AVL in mitigating Pb-induced ferroptosis, the levels of Fe2+ in brain tissues were assessed in this study. An evaluation of GPX4 expression was performed through Immunofluorescence and Western blot assays (Page 13-14).

  1. What criteria were used to choose the doses?

  Response:

Agree. Thanks for your comment. According to your suggestion, we have revised our manuscript. The procedure of Pb-induced nerve damage (1000 mg/L lead acetate solution in drinking water for 3 months) was performed as described (Wang et al., 2022). To eliminate various intake of Pb, drinking water and body weight of mice were monitored (at 17:00 everyday) throughout the study. Based on daily consumption of drinking solutions, the intakes of Pb were calculated. The daily intake of Pb was expressed as mg Pb/body weight/24 h. Our result showed that LD50 of Pb is 3214 mg/kg body weight. 1000 mg/L lead acetate (according to the daily intake of Pb) is about one-tenth of LD50 (Liu et al., 2011). Mice were also supplied with AVL 25 or 50 mg/kg intragastrically once daily. Based on previous studies, the dose of AVL, selected herein was sufficient to exert a Neuroprotective effect (Samant, and Gupta, 2020; Patel, 2021).

Liu, C.M., Zheng, G.H., Cheng, C., Sun, J.M., 2013. Quercetin Protects Mouse Brain against Lead-Induced Neurotoxicity J. Agric. Food Chem. 61, 7630−7635.

Liu, C.M., Ma, J.Q., Sun, Y.Z., 2011. Protective role of puerarin on lead-induced alterations of the hepatic glutathione antioxidant system and hyperlipidemia in rats. Food and Chemical Toxicology 49, 3119-3127.

Wang, W., Shi, F., Cui, J., Pang, S., Zheng, G., Zhang, Y., 2022. miR-378a-3p/549 SLC7A11 regulate ferroptosis in nerve injury induced by lead exposure.  Ecotoxicol. Environ. Saf. 239, 113639.

Patel, D.K. Medicinal importance of avicularin as potential anti-inflammatory agents for the treatment of liver disorders: Therapeutic assessment and biological importance in the medicine. Annals of Hepato-Biliary-Pancreatic Surgery 2021, 25, S296-S296.

Samant, N.P.; Gupta, G.L.  Avicularin attenuates memory impairment in rats with amyloid beta-induced alzheimer's disease. Neurotox. Res. 2022, 40(1), 140-153.

  1. Please, the figures must all have the same style.

  Response:

Agree. Thanks for your comment. According to your suggestion, we have revised our manuscript.

  1. Please, check the group name, why in some graphs the experimental group named "d" becomes "a"?

  Response:

Agree. Thanks for your comment. According to your suggestion, we have revised our Figures.

  1. Please, add significance in figures.

  Response:

Agree. Thanks for your comment. According to your suggestion, we have revised the Figures in manuscript. Values not sharing a common superscript letter (a–d) differ significantly at P < 0.05.

  1. What is the limitation of the study?.

  Response:

Agree. Thanks for your comment. According to your suggestion, we have revised our manuscript (Page 24). While this research noted that AVL could mitigate the neurotoxic effects induced by Pb in mice, the precise underlying mechanism remains ambiguous. Further exploration is warranted to understand the influence of AVL on other pivotal components involved in the pathogenesis of Pb-induced neurotoxicity.

  1. Please, add the paragraph of the discussion.

  Response:

Agree. Thanks for your comment. According to your suggestion, we have revised our manuscript. The paragraph of the discussion have added (Figure 8).

Thank you very much for the excellent and professional review of our manuscript!

Best regards!

Round 2

Reviewer 1 Report

The authors have improved the manuscript. I still have a few additional minor comments:

6. The soundness of the protein expression data presented in the manuscript (Figs. 2b, 3a, 4-6) cannot be properly evaluated because the original Western blot images provided as narrow membrane strips showing single polypeptide bands. Thus, the locations of these bands on the original membranes and the molecular weights of the polypeptides are unknown, which makes it impossible to identify the proteins. This is a major flaw. Therefore, the authors must provide the images of original untrimmed membranes showing molecular weight markers and sample names. The membrane images should be numbered according to the Figure numbers.

Response:

Agree. Thanks for your comment. According to your suggestion, we have revised data. We don’t have funding for the big PVDF membrane. We cut the corresponding size of the PVDF membrane for protein translocation based on the molecular weight of the target protein and accorded to the size of the marker protein in protein electrophoresis for protein translocation. We have indicated the molecular weights of the protein bands on the original membranes. The molecular weight markers also showed on the images of original untrimmed membranes. The membrane images had be numbered according to the Figure numbers.

Comments:

1. “The molecular weight markers also showed on the images of original untrimmed membranes.”

- The molecular weight of each standard marker protein should be indicated next to the corresponding polypeptide band. 

2. Line 191 (2.6. Western blot analysis):  Please provide details on the standard marker protein mixture used in the study, including the mixture name and catalog number.

3. Line 458: “In summary, the activation of the AMPK/Nrf2 pathway by AVL resulted in notable reductions in Pb-induced memory impairment, inflammation, ferroptosis, oxidative stress, and glucose metabolism disorder (Fig.8).”

- This sentence needs to be revised as follows: “In summary, the activation of the AMPK/Nrf2 pathway by AVL was associated with notable reductions...” because the authors did not present evidence that “the activation of the AMPK/Nrf2 pathway by AVL resulted in ...”. 

The authors have improved the manuscript. I still have a few additional minor comments:

6. The soundness of the protein expression data presented in the manuscript (Figs. 2b, 3a, 4-6) cannot be properly evaluated because the original Western blot images provided as narrow membrane strips showing single polypeptide bands. Thus, the locations of these bands on the original membranes and the molecular weights of the polypeptides are unknown, which makes it impossible to identify the proteins. This is a major flaw. Therefore, the authors must provide the images of original untrimmed membranes showing molecular weight markers and sample names. The membrane images should be numbered according to the Figure numbers.

Response:

Agree. Thanks for your comment. According to your suggestion, we have revised data. We don’t have funding for the big PVDF membrane. We cut the corresponding size of the PVDF membrane for protein translocation based on the molecular weight of the target protein and accorded to the size of the marker protein in protein electrophoresis for protein translocation. We have indicated the molecular weights of the protein bands on the original membranes. The molecular weight markers also showed on the images of original untrimmed membranes. The membrane images had be numbered according to the Figure numbers.

Comments:

1. “The molecular weight markers also showed on the images of original untrimmed membranes.”

- The molecular weight of each standard marker protein should be indicated next to the corresponding polypeptide band. 

2. Line 191 (2.6. Western blot analysis):  Please provide details on the standard marker protein mixture used in the study, including the mixture name and catalog number.

3. Line 458: “In summary, the activation of the AMPK/Nrf2 pathway by AVL resulted in notable reductions in Pb-induced memory impairment, inflammation, ferroptosis, oxidative stress, and glucose metabolism disorder (Fig.8).”

- This sentence needs to be revised as follows: “In summary, the activation of the AMPK/Nrf2 pathway by AVL was associated with notable reductions...” because the authors did not present evidence that “the activation of the AMPK/Nrf2 pathway by AVL resulted in ...”. 

Author Response

1. The molecular weight of each standard marker protein should be indicated next to the corresponding polypeptide band.

  Response:

 Agree, according to your suggestion, the molecular weight of each standard marker protein should be indicated next to the corresponding polypeptide band.

  1. Line 191 (2.6. Western blot analysis):  Please provide details on the standard marker protein mixture used in the study, including the mixture name and catalog number.

  Response:

 Agree. Thanks for your comment. According to your suggestion, we have revised our manuscript (Page 9).

  1. Line 458: “In summary, the activation of the AMPK/Nrf2 pathway by AVL resulted in notable reductions in Pb-induced memory impairment, inflammation, ferroptosis, oxidative stress, and glucose metabolism disorder (Fig.8).”- This sentence needs to be revised as follows: “In summary, the activation of the AMPK/Nrf2 pathway by AVL was associated with notable reductions...” because the authors did not present evidence that “the activation of the AMPK/Nrf2 pathway by AVL resulted in ...”.

  Response:

Agree. Thanks for your comment. According to your suggestion, we have revised our manuscript (Page 21).

Reviewer 2 Report

Very good demonstration of the biological activities of vicularin.

I agree with these WB. These are these WB which must be on the website as supplementary data.

Author Response

1. Very good demonstration of the biological activities of avicularin. I agree with these WB. These are these WB which must be on the website as supplementary data.

  Response:

      We would like to thank referees for comments. We thank the reviewers for their recommendation.

Reviewer 3 Report

no comment

no comment

Author Response

1. no comment

  Response:

      We would like to thank referees for comments. We thank the reviewers for their recommendation.

Thank you very much for the excellent and professional review of our manuscript!